# Multilinear Regression Analysis between Local Bioimpedance Spectroscopy and Fish Morphological Parameters

Vincent Kerzérho [1,*] , Florence Azaïs [1], Serge Bernard [1], Sylvain Bonhommeau [2], Blandine Brisset [2], Laurent De Knyff [1], Mohan Julien [2], Michel Renovell [1], Tristan Rouyer [3] , Claire Saraux [4] and Fabien Soulier [1]

1   LIRMM, Univ Montpellier, CNRS, 161 rue Ada, CEDEX 5, 34095 Montpellier, France
2   IFREMER/DOI, rue J. Bertho, 97822 Le Port, France
3   IFREMER, Univ Montpellier, MARBEC, Av. J. Monnet, 34203 Sète, France
4   IPHC, CNRS, UDS, 23 rue Becquerel, 67087 Strasbourg, France
*   Correspondence: vincent.kerzerho@lirmm.fr

**Abstract:** Repeated fish handling may cause stress, which biases experiments and so affects the results. In order to reduce this, the present study investigates the benefit of using bioimpedance analysis to estimate morphological parameters. Bioimpedance analysis is a non-lethal and integrative electrical measurement that can be used to estimate several kinds of physiological parameters and is used in medicine and ecological studies. In particular, bioimpedance can monitor the individual growth of fish, which is a prerequisite for most biological and ecological studies, as body size is one of the best predictors for numerous life history traits and ecological parameters. After a short review on the use of bioimpedance measurement in medicine and ecology, we illustrate the potential of bioimpedance spectroscopy, as opposed to single frequency measurement, for experimental studies on fish. Using a monolithic four-contact electrode and a cost-effective portable system, we conducted bioimpedance spectroscopy on 159 sardines. The association between the length, weight, and electrical parameters obtained at different frequencies from the bioimpedance spectroscopy was investigated. Our results show that accounting for more than one frequency substantially improves the prediction of length and weight. We conclude that bioimpedance could potentially be a powerful tool for monitoring fish growth in ecological studies.

**Keywords:** bioimpedance; spectroscopy; morphology; machine learning

## 1. Introduction

### 1.1. Context

A very general definition describes bioimpedance as being about the electrical properties of biological structures from the cell level to the entire body. More precisely, it is a measure of how well a biological structure impedes an electric current flow. This publication is using bioimpedance applied at fish body level. Usually, bioimpedance is measured by applying a known alternating current of milliamperes in a frequency range of tens or hundreds of kilohertz and acquiring the corresponding voltage drop via electrodes. In practice, the same two electrodes can be used for the current and voltage, but four electrodes can also be used, with two being used for the current and two for the voltage. The ratio of measured voltage-to-applied current yields the complex impedance value. The applied current flows through the body tissues made of cells and intra- and extracellular fluids, making the measured bioimpedance value dependent on the tissue composition. Consequently, a large amount of meaningful physiological information can be deduced from these bioimpedance values, which are used in various applications, such as skin water content, body composition, tissue ischemia monitoring, meat-quality assessment, etc. The number of bioimpedance applications is so vast that Geddes and Baker wrote: "The elegantly simple technique requires only the application of two or more electrodes, and it has been used successfully for many years to detect a remarkable variety of physiological events".

The correlation between the bioimpedance values and physiological parameters is more than evident. However, there may be more useful information contained in these bioimpedance values than just physiology—until today there has been no specific study. The original idea behind this paper is to investigate a possible correlation between bioimpedance values and morphological parameters. In other words, this paper explores the feasibility of estimating the length and weight of the entire body using bioimpedance measurement, regardless of the size of the measured part of the body. Indeed, we consider a general context, where the measurement is not necessarily taken from the whole body, even if we anticipate that the correlation will be more difficult to establish. In brief, the objective of the paper is to correlate a local bioimpedance measurement with global morphological parameters, such as length and weight.

Deducing both the physiological and morphological information from a simple and single measurement is of the greatest interest in many applications. For example, in this paper, we are applying it to experiments on fish. With this, a large range of characteristics need to be routinely monitored, such as a fish's morphological characteristics (size and weight) and physiological state (fat content and maturity). In relation to the physiological parameters, bioimpedance analysis is widely used due to several very interesting facets because it is a non-lethal technique that allows for the assessment of fat content or body composition. In comparison, the more common approaches, such as biochemical analyses, require sacrificing the fish or waiting until the experiment is over, which reduces the amount of data gathered. Concerning the morphological parameters, the length and weight measurements mandate the repeated handling of the fish, which can, in turn, introduce bias into the experiment's results (e.g., a decrease in growth through stress, as described by [1,2]). The development of a system allowing for the collection of such information with minimal fish handling is, therefore, of interest, as it protects the well-being of the studied animals, while increasing the quality of the results and/or allowing for the collection of new information (e.g., during the course of open sea migrations). In relation to the practical measurement, until now, this has been performed via a couple of electrodes located at the extremities of the fish: in the head and tail. Data are then normalized according to the fish size, which is precisely the sought-after unknown information. In order to decrease fish handling and make the measuring consistent, it would be interesting to use a single monolythic support with a fixed dimension and two electrodes. This support could easily be used to perform local bioimpedance measurements.

This paper raises two unanswered questions in relation to the application of bioimpedance analysis on the monitoring of key body characteristics of fish: (i) Is there a correlation between the bioimpedance measurements and global morphological parameters such as length and weight? (ii) Can a local, rather than a global (whole body), bioimpedance measurement provide relevant information? This paper does not provide a complete and definitive answer, as more experiments are necessary, but it does represent the first attempt at considering this new direction in bioimpedance applications.

In the remainder of the paper, Section 1.2 describes the different techniques that are typically used today for bioimpedance measurements. The basic principles are explained, and their applications in humans and fish are detailed together, along with the current electronic equipments. In Section 2, the method and materials used in our experiments are presented. Section 3 then analyzes and comments on the obtained results, followed by Section 4, which is the conclusion.

*1.2. Conventional Bioimpedance Analysis*

1.2.1. Bioimpedance Analysis Principle

Electrical impedance is an electrical parameter used for circuits that provides a measurement of the opposition of a circuit to the passage of electrical current. It is a complex number presented by the Equation (1).

$$Z(f) = R(f) + i\,X(f) \tag{1}$$

where $Z(f)$ is the impedance, $R(f)$ is the real part (also called resistance), $X(f)$ is the imaginary part (also called reactance), and $f$ is the frequency of the electrical signal.

The impedance concept can be extended to any conductive material, but with biological tissues, it is called bioimpedance. Because $U(f) = Z(f)xI(f)$, bioimpedance can be measured by generating an alternating current signal $I(f)$ at frequency $f$ through the tissue and measuring the induced voltage signal $U(f)$ or reversely by controlling the voltage $U(f)$ and measuring the induced current $I(f)$. Therefore, bioimpedance refers to the electrical property of a biological tissue [3], whose conductivity varies according to its composition. Indeed, the bioimpedance is determined by the water and lipid content of the sample under consideration. Intracellular and extracellular liquids contain ions. Due to these free ions (mainly Na+ and K+), extracellular and intracellular fluids are considered electrolytes, which means that they have the ability to conduct an electrical current in the presence of an external electric field [4]. In contrast, cell membranes and lipids act as an electrical capacitor. Consequently, impedances are due to extracellular water (ECW), intracellular water (ICW), and cell membrane impedance.

The tissue can be modeled by two impedances in parallel, as illustrated in Figure 1. For low frequencies (less than 50 kHz), the impedance of the membrane is very high, and the current, therefore, only flows through the extracellular fluids. In contrast, for high frequencies (above 50 kHz), the current flows through the extracellular and intracellular fluid and through the cell membrane [5].

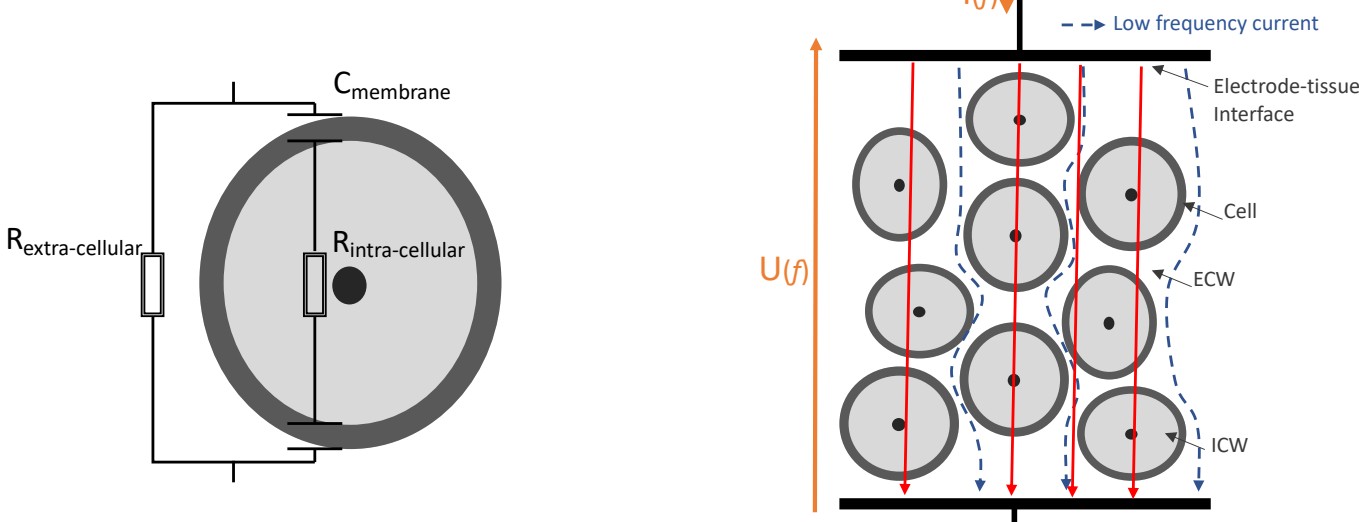

**Figure 1.** Intra and extra-cellular impedance. Depending on frequency, measured impedance is mainly related to extracellular water (ECW) or intracellular water (ICW).

From an experimental point of view, the bioimpedance measurement is performed with several electrodes placed on the sample and can be achieved with a variety of electrode configurations and numbers. The simplest configuration consists in using only two electrodes, implying that the stimulation and measurement are performed with the same contacts. This is usually called a two-point measurement configuration, as represented in Figure 2. In this case, the measurement includes three impedances in series: the bioimpedance of the tissue, which is the real target, but also the impedances of the interface between the two electrodes and the tissue, which interfere with the real target. Undeniably, the interface impedance can be much higher than the impedance of the tissue and can dominate the measurement. In addition, the interface impedance varies according to independent factors, such as the quality of the mechanical contacts and the biological evolution of the interface due, for example, to a growth of fibrotic tissues.

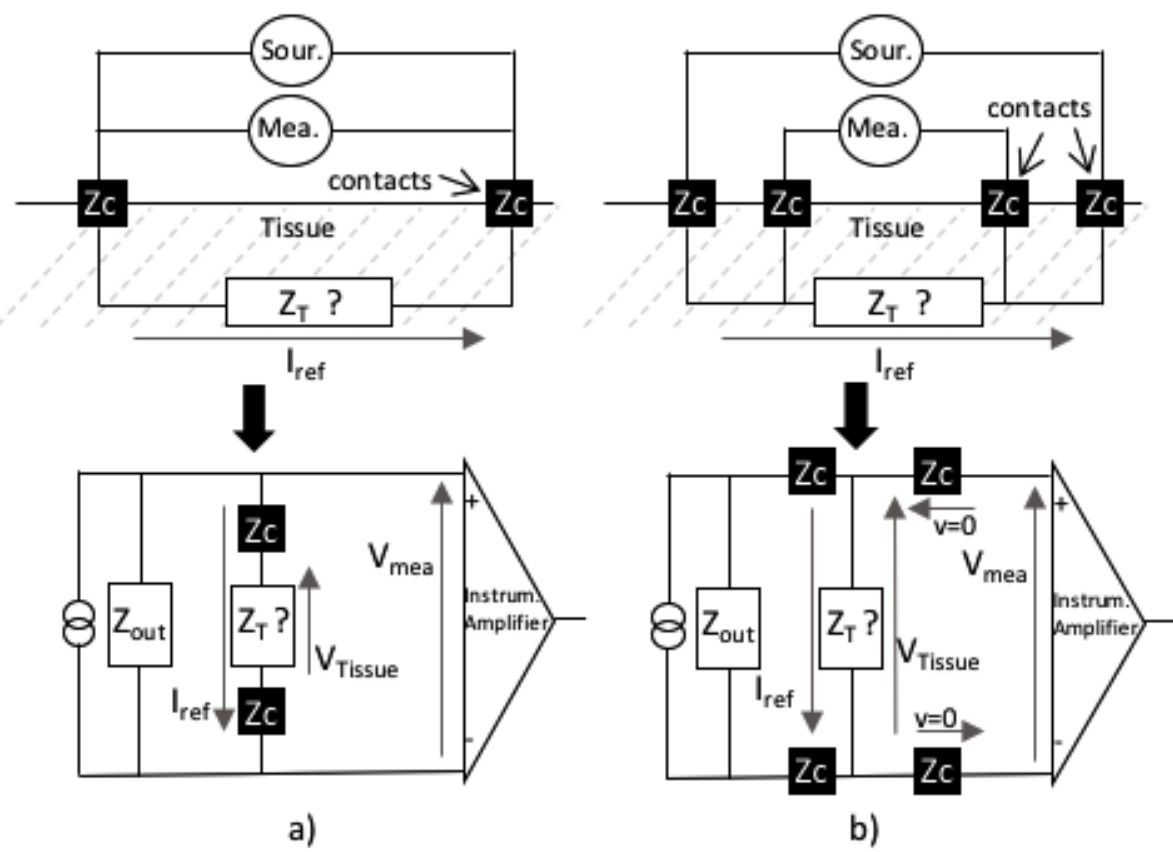

**Figure 2.** Measurement with (**a**) 2-point and (**b**) 4-point configurations.

An alternative is to use a more complex, but more efficient, configuration, i.e., two electrodes for stimulation plus two electrodes for measurement, usually referred to as a four-point measurement configuration. Since the measuring points do not draw current through their contacts, they do not affect the interface impedances.

### 1.2.2. Bioimpedance Analysis in Medicine

Bioimpedance measurements are used in medicine to establish the overall composition of an individual's body or body part [6]. The most commonly extracted parameters are total body water (TBW), extracellular fluid (ECL), fat mass (FM), and fat free mass (FFM). The bioimpedance measurement is used as a simpler and less expensive alternative to the DXA (dual-energy X-ray absorptiometry) reference measurement. It is usually performed on the entire body (leg-to-leg analysis) [7], but also for body segment composition analysis (arms, feet, legs, etc.) [8].

Considering this ability to estimate the overall composition of tissues, there are obviously many medical applications. The bioimpedance can be used to estimate the overall condition of the patient in post-operative follow-up or the general condition of an elderly person [9], for example. Bioimpedance techniques are also used for the detection, diagnosis, and monitoring of diseases [10], such as cancer [11]. Bioimpedance measurement can also be used to characterize biological components, such as blood [12].

Most often the bioimpedance measurement is performed at a single frequency of 50 kHz, for which the electric field lines pass through intra- and extracellular fluids, with a significant impact on cell membranes. The key parameter obtained from this measurement is usually considered to be the phase angle [9,13], which corresponds to the complex impedance argument, Equation (2). Because this measurement only considers one fre-

quency, it remains very sensitive to the measurement conditions and the physiological state of the patients [14].

$$Phase angle = Arg\{Z(f)\} = |\arctan(X(f)/R(f))| \qquad (2)$$

Another technique [15–17]. called BIVA (bioelectrical impedance vector analysis) uses vector graphics to analyze bioimpedance data with a normalized value of the impedance and additional information on patient characteristics: age, height, gender, etc. This technique also relies on a single 50 kHz frequency measurement, but an efficiency improvement is obtained by taking the global context of the analysis and, in particular, the specificity of the studied patient into account.

It is also possible to use multiple frequencies to obtain a bioimpedance signature. This improves the estimation's accuracy, but makes the signature analysis much more complex. Several approaches are used, such as measuring the impedance for a finite number of frequencies, called multi-frequency bioimpedance analysis (MF-BIA), or for a continuous frequency range, called electrical impedance spectroscopy (EIS) [18]. Many MF-BIA or EIS techniques use Cole–Cole modeling to extract information from the impedance measurement. The idea is to define the correlation between the Cole–Cole model parameters and the targeted biological parameters, in order to estimate these parameters [19–22].

Ever since the early use of bioimpedance analysis in the medical field to estimate the different components of the human body [23], the number of published papers per year has grown exponentially. On the one hand, this demonstrates the growing global interest in this practice. On the other, it shows that the technique remains unmastered and not completely standardized, maybe due to the excessive number of parameters [14,24].

Finally, it is worth noting that the different measurement protocols include a significant number of constraints, such as limited physical exercise before measurement, the immobility of the patient, absorption of a given quantity of water before measurement, etc., which are all incompatible with our context of a wild animal in its natural environment.

### 1.2.3. Bioimpedance for Fish

The use of bioimpedance analysis in fish is much more recent and less widespread. For example, the [25] paper presents one of the first reliable results using bioimpedance analysis to estimate the condition of the animal using a non-lethal technique. The overall idea was to assess if the bioimpedance measurement can provide additional information, compared to the classical morphological analyses on size, dry weight or wet weight [26,27], and the use of combined parameters, such as Fulton's condition [28], to estimate the animal's condition [29,30].

Various different studies have proposed using bioimpedance analysis to obtain a fish quality signature [31], for example, before and after freezing [32]. Obviously, in this case, the measurement is performed on dead fish to extract the signature.

Concerning the experimental conditions, the bioimpedance measurement is always carried out on the entire body, in order to obtain a general estimate of the individual. However, since the size of the individuals varies, the electrodes are consequently placed at varying distances, depending on the size of the fish [33]. With the value of the impedance being directly related to this inter-electrode distance, the measurement needs to be corrected. The proposed solution is to compensate this varying distance by "normalizing" the value by the size of the individual. The work of [33] presents the first correlation between a bioimpedance index and the morphological parameters of fish. As previously mentioned, the bioimpedance measurements are normalized by fish length, which makes the bioimpedance measurement for morphological parameter estimation useless. Other compensation techniques are also proposed for any parameter that could degrade the measurement, such as temperature, instrument accuracy, etc. [34,35].

In many medical applications, a single frequency of 50 kHz is used. In this study, we intend to fully explore the benefits of bioimpedance spectroscopy, which integrates several biological processes, as it provides information on the different constitutive elements of the

tissue: the cells and intra- and extracellular fluids. Moreover, in our protocol, the distance between the electrodes is constant, no matter the individual's size, so the resulting value is not normalized to the size. The proposed measurement is a local one performed on a few centimeters, while the researched information is global and concerns the whole fish.

1.2.4. Instrumentation for Bioimpedance Analysis on Fish

Concerning the in-situ measuring of fish, there are two different types of instrumentation. The first consists of complex and precise impedance spectroscopy instruments working on a wide range of frequencies and amplitudes, which are intended for use inside dedicated laboratories. The second consists of simpler and less efficient instruments, but ones that are portable and can be used in the field.

If we focus on the bioimpedance field measurement devices, there are very few dedicated commercial systems. There is undeniably a commercial portable bioimpedance spectroscopy measurement solution, but it is adapted to the global measurement of animals in a controlled environment (Impedivet), and it integrates the characterization models dedicated to mammals. The only equipment used for field measurements on fish is the QUANTUM IV (RJL system), which only allows measurements at a unique frequency of 50 kHz.

One of the objectives of this paper is to investigate the correlation between the bioimpedance measurement and morphological parameters on a large spectrum of frequencies from 0.3 kHz to 100 kHz.

## 2. Materials and Methods

### 2.1. Ethical Statement

All procedures were in accordance with the French and the EU legislation regarding animal experimentation (APAFIS, permission No.7097-2016093008412692).

### 2.2. Preliminary Remark

As explained in the introduction, the objective of this paper is to evaluate the possibility of using local bioimpedance measurements to derive the global morphological parameters of fish. Therefore, the evaluation presented in this paper was realized with fish placed on an experimentation table outside of the water.

Furthermore, we observed that commercial bioimpedance measurement instruments, as detailed in the previous section, may create limitations for our experiment. For this reason, we have designed and developed our own complete, small, portable, and battery-operated instrument, called a PIS (portable impedance spectroscope), which is able to perform bioimpedance measurements on a large frequency range of 0.3 kHz to 100 kHz.

### 2.3. Method

In this study, measurements were performed on 159 European sardines (Sardina pilchardus) captured in the French Mediterranean Sea (near Sète) and kept in experimental tanks at the IFREMER experimental platform in Palavas-les-Flots, France. A total of 40 sardines had been caught in March 2016, with a second batch of 119 pilchards in October 2016. After a month of acclimation in 5 m$^3$ outside tanks, fish were kept in smaller experimental tanks and fed twice a day with different quantities of aquaculture pellets of varying sizes (see [36] for more details).

In March 2017, the 159 Mediterranean sardines were anesthetized using benzocaine balneation at 140 ppm, and bioimpedance measurements were taken on all fish. The total handling time per fish for wet weight, fork length, and bioimpedance measurements was no more than 15 s.

In order to test for repeatability and the consistency of measurements taken with the instrument, measurements were repeated twice with two of our portable impedance spectroscopes (PIS), noted PIS1 and PIS2 below. In addition, for each PIS, the measurements

were again repeated twice with two different current levels: the first one with 100 µA and the second one with 400 µA. Finally, for each of the 159 fishes, the same procedure was used:

- First, the fish was anesthetized and both total length and weight (called here experimental weight Wexp and length Lexp) were measured.
- Second, a monolithic electrode of 4 cm long and 0.5 cm wide with two sets of needle electrodes, each consisting of a signal and detecting electrode, were inserted to a depth of 0.5 cm. The monolithic electrode was placed towards the back of the fish under the dorsal fin, which corresponds to the pterygiophores region.
- Third, a first PIS (PIS1) was connected to the electrode wire: A current of 100 µA was generated for 512 different frequencies, ranging from 0.3 Hz to 100 kHz, and the corresponding voltages were measured. Then, a second current of 400 µA was generated with the same 512 frequencies, and the corresponding voltages were measured. For each measurement, the PIS1 instrument provided the real part $R(f) = \Re\{Z(f)\}$ and the imaginary part $X(f) = \Im\{Z(f)\}$ of the corresponding impedance $Z(f)$.
- Fourth, the PIS1 was disconnected from the electrode wire, and a second spectroscope, the PIS2, was connected: again 512 measurements were performed with a current of 100 µA and 512 measurements with 400 µA. It is important to note that spectroscopes 1 and 2 were interchanged by disconnecting the spectroscope from the wire, but without moving the electrode. This part of the procedure helps evaluate the possible inaccuracy that could come from the instruments.

According to the literature [29,35], temperature affects bioimpedance measurement. To avoid this impact, we did the experiment in a lab with a controlled environmental temperature.

After the measurement procedure, the second step of the experiment was performed: electrical parameters derivation. From the measurement procedure, for each of the 159 fishes, we obtained two morphological data ($L_{exp}$ and $W_{exp}$), plus $4 \times 512$ electrical data with PIS1 and $4 \times 512$ data with PIS2: real part $\Re\{Z(f)\}$ and imaginary part $\Im\{Z(f)\}$ for I = 100 µA and for I = 400 µA measured for 512 different frequencies. From these $2 \times 4 \times 512$ electrical data, it is then possible to compute the $2 \times 6 \times 512$, following electrical parameters corresponding to the module Mo(f) of the impedance, the phase Ph(f) of the impedance, the equivalent serial resistance $R_s(f)$, the equivalent serial reactance $X_s(f)$, the equivalent parallel resistance $R_p(f)$, and the equivalent parallel reactance $X_p(f)$. These six electrical parameters, called $Epar_i$, are computed from $\Re\{Z(f)\}$ and $\Im\{Z(f)\}$ with the following equations:

$$Epar_1(f) = R_S(f) = \Re\{Z(f)\} \tag{3}$$

$$Epar_2(f) = X_S(f) = \Im\{Z(f)\} \tag{4}$$

$$Epar_3(f) = R_P(f) = R_S(f) + \frac{X_S(f)^2}{R_S(f)^2} \tag{5}$$

$$Epar_4(f) = X_P(f) = X_S(f) + \frac{R_S(f)^2}{X_S(f)^2} \tag{6}$$

$$Epar_5(f) = Mo(f) = |Z(f)| \tag{7}$$

$$Epar_6(f) = Ph(f) = arctan\left(\frac{R_S(f)}{X_S(f)}\right) \cdot \frac{180}{\pi} \tag{8}$$

$R_s(f)$ and $X_s(f)$ are, respectively, equal to the previous R(f) and X(f). Undeniably, Z(f) is the impedance of a resistive element and a capacitive one when in series. We added the s to R(f) and X(f) to distinguish them from $R_p(f)$ and $X_p(f)$, which are the real and imaginary parts of an electrical model, considering a resistive element and a capacitive one in parallel.

Finally, the third step of the experiment was performed: the statistical analysis. We first analyzed the individual characteristics and distributions, in terms of the mean and standard deviation of the morphological parameters and the electrical parameters. In the distributions of the electrical parameters, we noted a good homogeneity and coherence for a range of frequencies from 1 kHz to 100 kHz, which was used in the remainder of the analysis. The obtained data were then used to analyze the correlation between the two morphological parameters, $W_{exp}$ and $L_{exp}$, and the six electrical parameters $Epar_i$ at given frequencies (see below). For this purpose, a typical approach was used based on multi-linear regression analysis. Multiple linear regression attempts to model the relationship between two or more explanatory variables and a response variable by fitting a linear equation to observed data. Every value of the independent variable is associated with a value of the dependent variable. Taking into consideration the six electrical parameters at the 512 measurement frequencies and the 159 fishes, in order to limit the number of predictors, we used only 66 of them, six electrical parameters for 11 frequencies. Consequently, we selected 11 values regularly spaced out in the range defined above for consistency purposes, i.e., 1 kHz to 100 kHz. The selected frequencies are the following:

$f_1 = 0.885$ kHz $f_2 = 10.635$ kHz $f_3 = 20.384$ kHz $f_4 = 30.133$ kHz $f_5 = 39.883$ kHz $f_6 = 49.632$ kHz $f_7 = 59.381$ kHz $f_8 = 69.131$ kHz $f_9 = 78.880$ kHz $f_{10} = 88.629$ kHz $f_{11} = 98.379$ kHz

Considering that $N_{var}$ is the number of variables for the multi-linear regression analysis, the general linear equations for $W_{est}$ and $L_{est}$ are the following:

$$W_{est} = c_0 + \sum_{k=1}^{N_{var}} c_k Epar_{i_k}(f_{j_k}) \text{ and } L_{est} = d_0 + \sum_{k=1}^{N_{var}} d_k Epar_{m_k}(f_{n_k}) \tag{9}$$

with

$$i_k, m_k \in \{1, 2, 3, 4, 5, 6\} \text{ and } j_k, n_k \in \{1, 2, 3, 4, 5, 6, 7, 8, 9, 10, 11\}$$

Example of a single variable regression $N_{var}=1$, with $i_1 = 5$, $j_1 = 2$:

$$W_{est} = c_0 + c_1 Epar_5(f_2) \tag{10}$$

Example of a two variables regression $N_{var}=2$, with $m_1 = 6$, $n_1 = 11$, $m_2 = 1$, $n_2 = 3$:

$$L_{est} = d_0 + d_1 Epar_6(f_{11}) + d_2 Epar_1(f_3) \tag{11}$$

In the above example for Lest, it is interesting to note that multiple Epar and multiple frequencies (ME-MF) are combined corresponding to the most general case. However, it is also possible to consider special cases, such as multiple Epar with a single frequency (ME-SF), as well as single Epar with multiple frequencies (SE-MF). Examples:

$$ME - SF => W_{est} = c_0 + c_1 Epar_5(f_9) + c_2 Epar_1(f_9) \tag{12}$$

$$SE - MF => W_{est} = c_0 + c_1 Epar_3(f_1) + c_2 Epar_3(f_8) \tag{13}$$

By using all the possible combinations of the electrical parameters and the frequencies, it is possible to compute a large number of different linear equations for $W_{est}$ and $L_{est}$. This number of equations $N_{equ}$ can be easily calculated in the following way:

- First, with six electrical parameters and 11 frequencies, the number of different terms $Epar_i(f_j)$ is $6 \times 11 = 66$.

- Second, considering a regression with $N_{var}$ variable, the linear equations are made of $N_{var}$ terms. So, the number of different equations corresponds to the number of combinations with no repetition, i.e., 66 choose $N_{var}$:

$$N_{equ} = \frac{66!}{N_{var}!(66 - N_{var})!} \tag{14}$$

As examples, a single variable regression $N_{var} = 1$ gives $N_{equ} = 66$, a two variables regression $N_{var} = 2$ gives $N_{equ} = 2145$, and a three variables regression $N_{var} = 3$ gives $N_{equ} = 137{,}280$. These numbers mean that thousands of equations have to be computed, leading to a significant CPU time but one that is still affordable. This justifies the selection of a limited number of 11 frequencies. However, it worth mentioning that the number of equations with the 512 frequencies and six $Epar_i$ for a three variables regression would be $1.4 \times 10^{10}$, which is obviously not affordable!

Looking at these thousands of equations of West and Lest obtained with 11 frequencies and six electrical parameters, the final objective is obviously to find the equation corresponding to the best correlation between $W_{exp}$ and $W_{est}$ and also $L_{exp}$ and $L_{est}$. For this purpose, for each equation, the characteristics $W_{est}$ versus $W_{exp}$ and $L_{est}$ versus $L_{exp}$ are created with the 159 points corresponding to the 159 fish. Additionally, for each one, the Pearson correlation coefficient, also referred to as Pearson's r, is computed. This coefficient reflects the quality of the correlation:

$$r = \frac{cov(W_{est}, W_{exp})}{\sigma_{W_{est}} \sigma_{W_{exp}}} \; and \; r = \frac{cov(L_{est}, L_{exp})}{\sigma_{L_{est}} \sigma_{L_{exp}}} \tag{15}$$

where cov is the covariance and $\sigma$ is the standard deviation.

It is then possible to rank the models in descending order of the Pearson coefficient, and of course, only the top ones are considered.

Another usually used ranking criteria is the Akaike information criterion (AIC), which is a good complementary to Pearson's coefficient, as it integrates a penalty related to the number of variables used in the model. It helps deal with the risks of overfitting and underfitting. So, the AIC provides a means for equation selection, knowing that the best one corresponds to the lower value of the AIC. In our case study, based on a small sample size, we have chosen to the use the corrected $AIC_c$, given by the following equation.

$$AIC_c = 2N_{var} + N_{fish}.ln\left(\frac{\sum_{k=1}^{N_{fish}} (W_{estk} - W_{expk})}{N_{fish}}\right) + \frac{2N_{var}(N_{var}+1)}{N_{fish} - N_{var} - 1} \tag{16}$$

and

$$AIC_c = 2N_{var} + N_{fish}.ln\left(\frac{\sum_{k=1}^{N_{fish}} (L_{estk} - L_{expk})}{N_{fish}}\right) + \frac{2N_{var}(N_{var}+1)}{N_{fish} - N_{var} - 1} \tag{17}$$

where $N_{fish}$ is the number of samples, i.e., 159 fishes.

## 3. Results

### 3.1. Individual Parameter Analysis

Table 1 gives the mean, standard deviation, and coefficient of variation of the two distributions. It is interesting to note that the length of the electrode (4 cm) used corresponds to one third of the fish's size. For detailed information, the weight and length distributions are provided in Figure S1.

**Table 1.** Characteristics of the morphological parameter distributions.

|  | **Weight** | **Length** |
|---|---|---|
| Mean Value | 21.4 | 13.4 |
| Standard Deviation | 12.4 | 1.8 |
| Coefficient of Variation | 57.9% | 13.7% |

In order to estimate the possible dispersion of the electrical parameters due to the measurement conditions, we plotted in Figure 3 the serial resistance and reactance $R_s(f)$ and $X_s(f)$ versus frequency characteristics for the four conditions: two currents (100 μA and 400 μA) and two PIS (PIS1 and PIS2). Ideally, without dispersion, the four characteristics should be equal. However, we obviously observed some differences, depending on the current level and the PIS. Dispersion is not easy to analyze for these characteristics and, by selecting $R_s(f)$ and $X_s(f)$, as an example, the mean values are given in Figure 4 for the four conditions, which leads to the following comments:

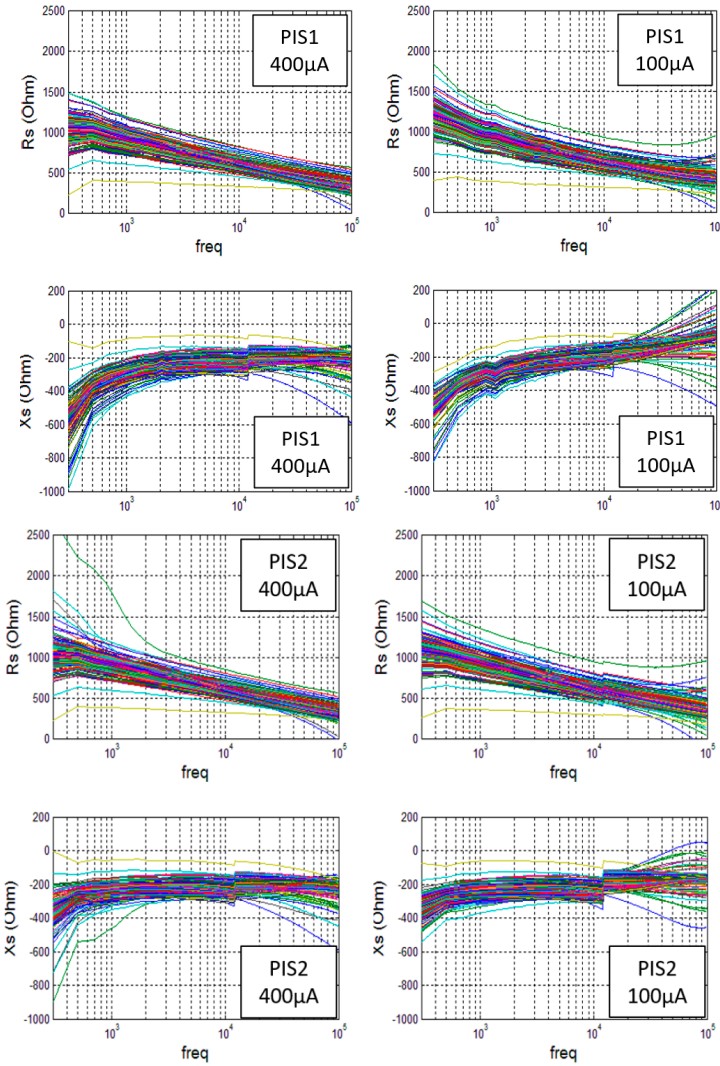

**Figure 3.** Serial resistance and reactance versus frequency.

- Firstly, a good consistency may be noted for $R_s(f)$ in a range from 1 kHz and 50 kHz. In the low frequency domain, the two spectroscopes provide very similar results when using the 400 μA current, but the PIS1 gives higher values than the PIS2 when a low

current of 100 μA is used. A similar inconsistency can be observed for frequencies higher than 50 kHz.

- Secondly, a good consistency may be noted for $X_s(f)$ in a very limited range of 2 kHz to 6 kHz. In the low frequency domain, a single PIS gives consistent results for different currents, while two different PIS give very different values. In the high frequency domain, the two different PIS give consistent results but only for a current of 400 μA.

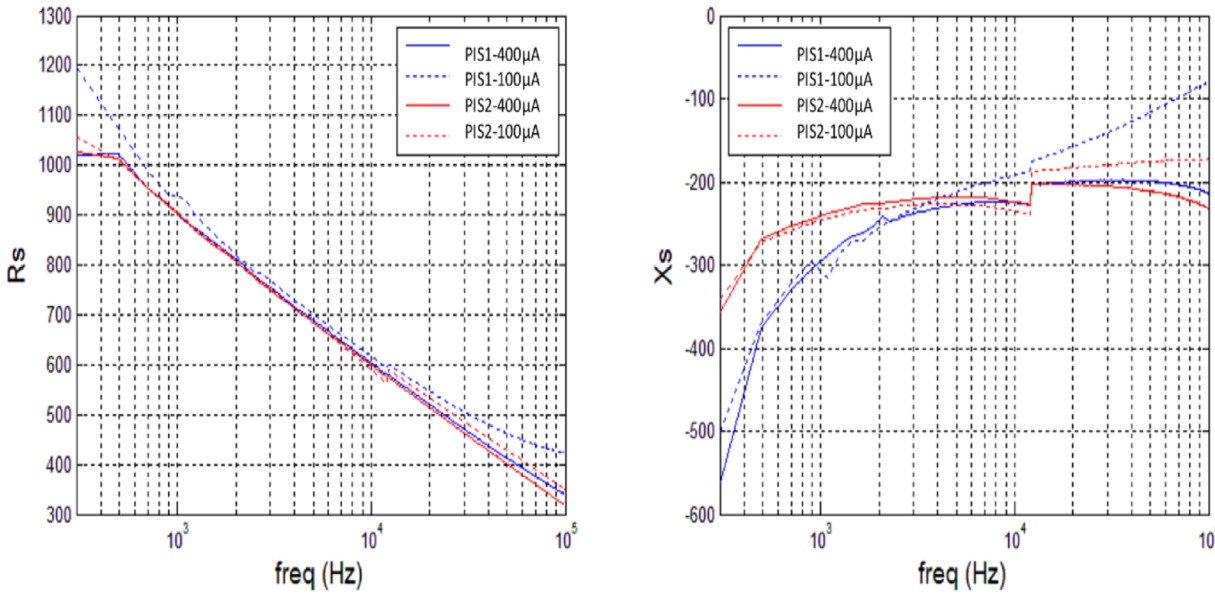

**Figure 4.** Mean value of $R_s(f)$ and $X_s(f)$ versus frequency.

In brief, these results clearly indicate that a good consistency is obtained with the two PIS in a wide range of 2 kHz to 100 kHz that is conditional on using a current of 400 μA. This observation justifies the selection of the 11 frequencies $f_i$ in this range. Measurements are very sensitive when the current is too low, so only the high current is considered in the following.

*3.2. One to Five-Parameter Multilinear Regression Models*

Bioimpedance spectroscopy makes the training of multi-parameter models possible. Despite any proof of linear relationships between bioimpedance and morphological parameters, as the first step in an exploratory analysis, we trained multilinear regression models.

Six electrical parameters $Epar_i$ were computed at each of the 11 discrete frequencies $f_j$ of the bioimpedance spectrocopy for weight and length. This resulted in 66 different correlation functions for $W_{est}$ and 66 different correlation functions for $L_{est}$ being computed. These 132 models were trained for the two measurement devices, PIS1 and PIS2.

We first observed a clear consistency between the measurements coming from PIS1 and PIS2, namely that the best correlations were obtained with the same electrical parameter at the same frequency. The consistency between PIS1 and PIS2 illustrated here was found for all the measurements, and so only the measurements with PIS1 are given in the remainder of the paper. Secondly, we observed that length provided the best results, with an excellent correlation coefficient of over 0.8. For more detailed results, the top six Pearson's r values with a single variable regression for the four cases are given in Table S1.

The best correlation functions have been computed with their corresponding correlation coefficients and corresponding corrected AIC for 1, 2, 3, 4, and 5 parameters. Figure 5 gives the Pearson's r and AICc versus $N_{var}$ characteristics for the weight and length. Three diferent cases are considered for the choice of variable: ME-SF (multiple

electrical parameters—single frequency), SE-MF (single electrical parameter—multiple frequencies), and ME-MF (multiple electrical parameters—multiple frequencies).

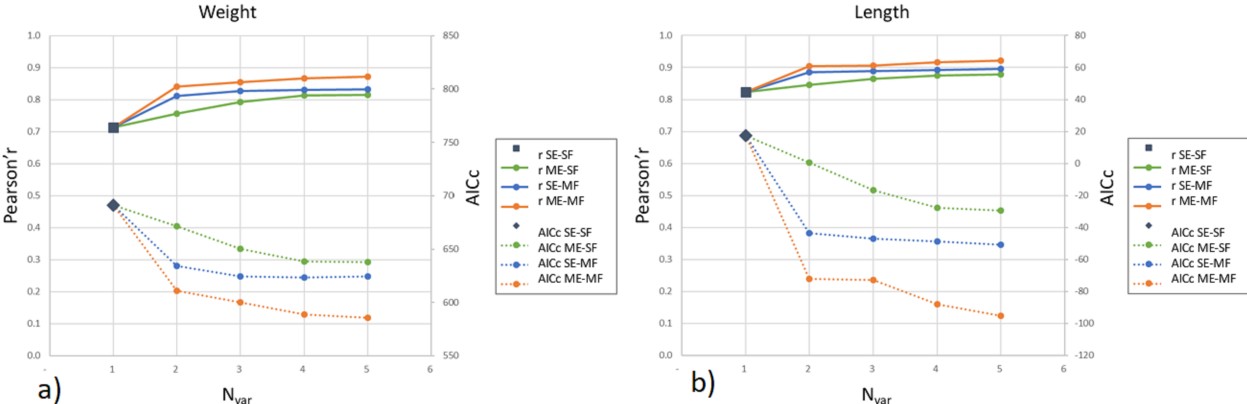

**Figure 5.** (**a**) Pearson's r and $AIC_c$ of the best correlation equations between observed and estimated weight for various number of parameter models (Nvar = 1.5) and for four different ways to select the model features (SE-SF, ME-SF, SE-MF- ME-MF), (**b**) Pearson's r and $AIC_c$ of the best correlation equations between observed and estimated length for various number of parameter models (Nvar = 1.5) and for four different ways to select the model features (SE-SF, ME-SF, SE-MF- ME-MF).

These characteristics clearly demonstrate that (i) the ME-MF approach always provides the best results, and (ii) the multi-variable regression gives much better results than a single variable, since the increase in r and the decrease in $AIC_c$ are very significant when going from one to two variables. However, the improvements are still present but less significant when going to more than three variables. The decision on the number of variables is a trade off between accuracy and the complexity of the computation, but a solution with around three variables seems to be a reasonable choice.

The detailed top correlation results for two to five-variable regression models are, respectively, given in Tables S2–S4.

## 4. Discussion and Conclusions

A statistical study of 159 living European sardines was performed by measuring the bioimpedance of each individual, together with its morphological parameters. For each of the 159 sardines, the bioimpedance was measured at 512 different frequencies, ranging from 0.3 kHz to 100 kHz. These measurements were repeated four times, for 100 µA and 400 µA, in order to evaluate the impact of the current level and with two different instruments to estimate the possible inaccuracy of the instruments. As a first observation, it is noteable that a high current of 400 µA gave more consistent results, while both instruments provided consistent measurements.

Six different electrical parameters $Epar_i$ were derived from the impedance measurements. Among the 512, a sample of 11 frequencies $f_j$ that were regularly spaced out were selected, in order to limit the computation complexity. It was verified that a different set of 11 frequencies does not modify the conclusion of the study. The multiple linear regression analysis was consequently performed on the two morphological parameters and the six electrical parameters considered at the 11 selected frequencies. For the computation of the correlation functions, single variable (one $Epar_i(f_j)$) and multiple variable (two, three, four or five $Epar_i(f_j)$) regression analysis were considered. For the multiple variable regression analysis, three different cases were compared: multiple $Epar_i$ at the same frequency (ME-SF), single $Epar_i$ at different frequencies (SE-MF), and multiple $Epar_i$ at different frequencies. In any case, the Pearson's r corresponding to the obtained correlation function was computed in order to clearly evaluate the quality of the correlation.

First of all, it is very important to note that a wide range of values, from very low to very high, was found for the Pearson's r coefficient, depending on the number of

variables, the selected electrical parameters, and the selected frequencies. This clearly indicates that both parameters and frequencies must be carefully selected to obtain the best correlation functions.

It is clear that the results obtained with multiple frequencies are far better than those with a single frequency. This important point implies that instruments that are limited to a single frequency are completely inadequate for this purpose. In addition, instruments with a fixed single frequency of 50 kHz do not permit the acquisition of the best correlation function in any case, and this is exacerbated when looking for the length.

Finally, looking at the best results, i.e., the ME-MF case with five variables, we obtained Pearson's r coefficients of 0.87 for weight and 0.92 for length, which represent excellent results. Considering not just the best case, but all the results, the best equations gave correlation coefficients of over 0.8. These levels of correlation are very positive and indicate that good and reliable correlations can be obtained between local bioimpedance measurements and fish morphological parameters, such as weight and length.

This experiment shows that a bioimpedance measurement of fish muscle provides local information on the muscle that is related to its length. It is well-known that the growth of the fish muscle follows two processes called hyperplasia and hypertrophia [37]. Hyperplasia corresponds to the recruitment of new fibers while hypertrophia corresponds to the enlargement of existing fibers. Plus, according to [37], there are strong correlations between fiber characteristics (average fiber size and number of fibers) and the length of the fish in the case of white seabass. Moreover, there is a good correlation between the percentage of hyperplastic fibers and the length of the fish. This information is related to white seabass, but it can be speculated that a similar correlation exists for Mediterranean pilchard. The competition between hyperplasia and hypertrophia vary with species, age and growing conditions. The impact of muscle growth on the relation between bioimpedance and morphology will be studied in complementary experiments with seabass, bluefin tuna, etc.

As a side note, these are preliminary results that highlight the benefit of bioimpedance spectroscopy for single-frequency bioimpedance. Good correlation coefficients do not guarantee estimations that are as precise as regular techniques. However, the main purpose of bioimpedance is not to replace regular techniques in any situation. For instance, it can be useful if the bioimpedance measurement system is integrated in a device attached to the animal for health monitoring. Such usage is very convenient for aquaculture and biologging applications. Biologging [38,39] is a research domain focused on the data collection of wild animals and their physical environment using electronic tags attached to the animals. Morphological parameter estimation using bioimpedance costs nothing if the bioimpedance measurement has already been planned for other health parameter estimations. It can become mandatory if the handling time is strictly limited for fish welfare or experiments on a large number of fishes.

Nevertheless, there is a need for additional research work for morphological parameter estimation that uses bioimpedance. It will be necessary to collect additional data for accurate model training and to look for the most suitable model type. In addition, it will need to be validated for various species. Furthermore, bioimpedance measurements are known to be sensitive to temperature. In order to characterize this impact in our case study and to establish a correcting procedure, we expect to run experiments in controlled conditions.

**Supplementary Materials:** The following supporting information can be downloaded at: https://www.mdpi.com/article/10.3390/fishes8020088/s1, Figure S1: Experimental weight and length distribution; Table S1: Top 6 Pearson'r with a single variable regression; Table S2: Top 6 Pearson'r with a two variable regression; Table S3: Best correlation equations with 3 variables; Table S4: Best correlation equations with 4 and 5 variables.

**Author Contributions:** All authors contributed to this study. Conceptualization, V.K., S.B. (Serge Bernard), S.B. (Sylvain Bonhommeau) and T.R.; methodology, V.K., C.S. and L.D.K.; data curation, F.A.; formal analysis, F.A.; writing—original draft preparation, M.R.; writing—review and editing, V.K., C.S., T.R., S.B. (Sylvain Bonhommeau) and M.J.; funding acquisition, V.K., S.B. (Sylvain Bonhommeau), S.B. (Sylvain Bonhommeau) and T.R. All authors have read and agreed to the published version of the manuscript.

**Funding:** This research was part of the MERLIN-POPSTAR project, funded by French Research Institute for Exploitation of the Sea (IFREMER), and in the framework of the FISHnCHIP project, funded by the European Maritime and Fisheries Fund (EMFF).

**Institutional Review Board Statement:** All procedures were in accordance with the French and the EU legislation regarding animal experimentation (APAFIS, permission No.7097-2016093008412692).

**Informed Consent Statement:** Not applicable.

**Data Availability Statement:** data available upon request

**Acknowledgments:** This research was supported by IFREMER aquaculture station at Palavas-les-Flots.

**Conflicts of Interest:** The authors declare no conflict of interest.

## Abbreviations

The following abbreviations are used in this manuscript:

| | |
|---|---|
| AIC | Akaike Information Criterion |
| BIVA | Bioelectrical Impedance Vector Analysis |
| ECL | Extracellular Liquid |
| ECW | Extracellular Water |
| EIS | Electrical Impedance Spectroscopy |
| EMFF | European Maritime and Fisheries Fund |
| DXA | Dual-Energy X-ray absorptiometry |
| FFM | Fat Free Mass |
| FM | Fat Mass |
| ICW | Intracellular Water |
| ME-SF | Multiple Epar Single Frequency |
| ME-MF | Multiple Epar Multiple Frequencies |
| MF-BIA | Multiple Frequencies BioImpedance Analysis |
| PIS | Portable Impedance Spectroscope |
| SE-MF | Single Epar Multiple Frequencies |
| TBW | Total Body Water |

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
