# Peer review of "Multilinear Regression Analysis between Local Bioimpedance Spectroscopy and Fish Morphological Parameters"

_fishes, doi:10.3390/fishes8020088_

Round 1

Reviewer 1 Report

This paper evaluates the use of bioelectrical impedance to predict fish length and weight.  Of particular interest in the paper is the use of a 4 contact electrode and using multiple currents and frequencies.  I think there is interesting data within this paper, but the manuscript and analysis would need substantial revision before it could be published.  In particular, I think the statistical analysis is incomplete and in some cases, not appropriate for the questions be asked.  Specific comments are below.

Abstract-

The abstract is not that focused on the importance and results from this study.  A large portion of the abstract is referencing another paper.  Most journals do not allow for a citation within the abstract.

Line 4 should read “several pieces of information” or something similar

Throughout the paper, the units for measures are not provided—for example, line 143 should be 50 kHz, line 187 should be 50 kHz, etc

Line 187 march should be March

line 234 Is this correct, I’m not sure I could measure a fish length in 6 seconds, much less length, weight, and the whole suite of BIA measures that were taken?

Line 252-253 these abbreviations are different than the ones provided on P. 3?

Line 328 (and throughout) figure should be Figure (or in some cases table should be Table)

Table 1 I’m not sure this is useful for two reason—1) the size distribution of fish is given in Fig. 3, and that is more useful than the mean size of fish used in the study and 2) given the bi-modal distribution, the mean is not that useful of a metric

Line 332 delete “it is assumed to be due to differing feeding conditions of fishes.”

I would say one of the biggest issues with the manuscript is the reliance on correlations to evaluate models.  Since the BIA measures were used to predict weight and length, the regression results and not correlations are the most useful statistics to report (p-values, r or R^2 from regression, RSS).  In turn, since AIC is being used, all models should be considered together, not separately as single, 2 factor, 3 factor, etc models.  This would greatly reduce the text within the results section as well.  Then the AIC results should provide the most parsimonious models, not a simple examination of the strength of the linear relationship between BIA measures and weight or length (i.e., correlation).  Then, the next step is to determine if the predicted weight or length is a good predictor of actual length or weight, using regression.  Evaluate the slope and intercept and see if they are 1 and 0 respectively, or if there is bias in the prediction.  The variability around the regression line is also important to evaluate the usefulness of the predictions.  As written, there is little or no evaluation whether the predicted length and weight are biased or not.  Fig. 6 does provide a glimpse into this question, and based on these two “good” models, I would say there is extensive bias in the predicted values, and the relationships do not even appear to be linear. 

Line 398 Here and elsewhere, it would be clearer to use kHz or Hz, but not both

Line 514 I don’t think you have shown that reliable estimates of length and weight can be derived from impedance measures.  Also, consider that only a small amount of error in estimates might be very important in experimental studies where an individual is measured multiple times over the study.  Thus, for this technique to be useful in some situations, it would have to provide very precise and accurate estimates of length and weight.

Line 529 this discussion is not that relevant to the objectives and could be deleted

Sample availability statement does not make sense

Not all abbreviations used in paper are shown in Abbreviations

Figure 1 Define abbreviations used in figure

Figure 3 -axis labels need units (grams, cm)

Fig. 4 second row missing x-axis labels

Fig. 6 Need units on x and y axis; define what the line is—it appears to be the 1:1 relationship line; the regression line through the points should be shown too so the relationship/bias can be evaluated

Reviewer 2 Report

I started reading this manuscript with the understanding that this is an article of original research, but now know that it is intended for a special issue.  I am not aware of the format of that special issue\, so my comments are in the context of expecting a research article.

I believe that the research described has some value, especially the development of a cost-effective portable system for bioimpedance spectroscopy.  That should be in the Abstract of the paper.  The format of the paper is such that it reads like some sort of hybrid manuscript between a review paper (with some new data) and a primary research article.  There is so much background information to wade through that it really detracts from the primary message - there is a correlation between bioimpedance and length/weight.  Example - if the manuscript doesn't address body condition, fat content, etc., why bring this up in humans?  The authors bring up the fact that this has been done before with a 2-pair electrode system, and they are using a 4-probe system, but this 'improvement' isn't discussed further (unless it is implicit in the regression models).

I believe that this manuscript could and should be reformatted as a research article, shortened considerably, and refocused towards the initial goal - development of a four probe system for estimating biological parameters.

 my comments before I stopped editing the language:

Kerzerho et al.

An attempt at correlating local bioimpedance measurements with fish morphological development

Comments:

Title:  suggest removing ‘An attempt’

L4:  that can provide several pieces of information from a single measurement

L6:  predictors

L7-11:  Suggest removing “A study (Willis et al. 2008, Fisheries research 93: 64-71) has presented a first correlation between bioimpedance index and length and weight of fishes. The bioimpedance measurements were done using two pairs of electrode implanted at the extremities of the fish and consequently the measurements were normalized by the length of the fish. In addition the bioimpedance has been measured at only one frequency”

Belongs in the Introduction, and the study should stand on its own merits rather than erecting a straw man.

L11-12:  “As a consequence we made an experiment using a monolithic 4-contact electrode for bioimpedance spectroscopy measurements.”  Suggest:  We designed an experiment using a monolithic 4-contact electrode for bioimpedance spectroscopy measurements to test the relationship between bioimpedance and morphological characters.

This manuscript needs some editorial attention to the English language and syntax.  I am going to stop correcting and just pay attention to the science.

Reviewer 3 Report

The manuscript (MS) was designed to calibrate electrical impedance for weigh and length of targeted fish species. It is very interesting MS, but focused on highly limited morphometrical estimation. 

Wholly reading, the MS was well organized and written. As presented in Fig. 6 for single variable (there is high intercept between estimated and experimental length, or weigh. The intercept could be expected to be near zero for the best fit), results of 1, 2, 3, and more variable model should be justified to draw experimental weight vs estimated weigh, and for length as well for better understating, rather than the best correlation. 

The MS could be accepted for a publication in Fishes after the justification aforementioned

Round 2

Reviewer 1 Report

The revision is a more concise and improved version of the manuscript.  I only had a couple questions yet to address.

1)      The introduction seems to indicate that an objective was to find if there was a relationship between BIA measures and body characteristics (line 69).  This objective is not as well explored as the as the objective to explore the role of different frequencies (line 206).  One way to further explore how useful this methodology might be is to show results from the best models (line 420) showing the observed characteristics as a function of the experimental results.  The importance of using multiple frequencies is more meaningful if you show the instrument can actually provide reliable predictions.

2)      For the AICc results, shown on Figure 5, were all AICc values determined when considering all models examined (1, 2, 3, 4, 5 parameter) together, or just within each set of models for a specific parameter number?  For the AICc to be useful, all models with all numbers (1 through 5) of parameters need to be considered together, and this will provide the most parsimonious model, with the most parsimonious number of parameters.  Otherwise, it is just providing the most parsimonious model for a given number of parameters. 

3)      Fig. 5 legend needs to describe what relationship the correlations are for (ie between Lexp and Lobs or Wexp and Wobs, and also if AICc is a comparison within each Nvar, which is not that useful, or among all models across varying levels of Nvar.

Reviewer 2 Report

The authors have done a good job of addressing previous comments.  I commend them for the reformat and shortening of the manuscript.  The manuscript does need some attention to English syntax, etc., but its minor.
